# A New Species of *Spelaeometra* Polhemus & Ferreira, 2018 (Hemiptera: Heteroptera: Hydrometridae) from a Hotspot of Troglobites in Brazil, Serra do Ramalho Karst Area [note 1]

**DOI:** 10.3390/ani13203199

**Published:** 2023-10-13

**Authors:** Isabelle R. S. Cordeiro, Maria Elina Bichuette, Felipe F. F. Moreira

**Affiliations:** 1Laboratório de Entomologia, Instituto Oswaldo Cruz, Fundação Oswaldo Cruz, Rio de Janeiro 21045-900, Brazil; 2Laboratório de Estudos Subterrâneos, Departamento de Ecologia e Biologia Evolutiva, Universidade Federal de São Carlos, São Carlos 13565-905, Brazil; lina.cave@gmail.com

**Keywords:** caves, semiaquatic bugs, taxonomy, troglomorphism

## Abstract

**Simple Summary:**

Water measurers (Hemiptera: Heteroptera: Hydrometridae) can be found mainly on marginal vegetation or on plants on the surface of freshwater bodies. Recently, two monotypic genera restricted to subterranean habitats were described from Brazil, namely *Cephalometra* Polhemus & Ferreira, 2018 and *Spelaeometra* Polhemus & Ferreira, 2018. Here, a second species of *Spelaeometra* is described based on material collected in a hotspot of subterranean animals in the Serra do Ramalho karst area, Brazil.

**Abstract:**

*Spelaeometra* Polhemus & Ferreira, 2018 (Hemiptera: Heteroptera: Hydrometridae) is a monotypic and troglobitic genus, recently described based on material collected in the state of Minas Gerais, Brazil. From specimens collected in the Gruna do Enfurnado Cave in the Serra do Ramalho karst area, municipality of Coribe, state of Bahia, Brazil, we describe *Spelaeometra hypogea* Cordeiro & Moreira, sp. nov. and compare it with *Spelaeometra gruta* Polhemus & Ferreira, 2018. The new species is diagnosed by the general body color being pale-yellow to light-brown; antennal segments I and II being subequal in length; the reduced gular lobe, not covering any labial segment; the male proctiger without apical projections; and the male paramere with convex dorsal margin after the curvature, with a narrow hook-like apex.

## 1. Introduction

Semiaquatic bugs (Hemiptera: Heteroptera: Gerromorpha) are a key component of the environments they occupy and the most successful animal group at the air–water interface [1]. Most of its representatives live on the surface or on the margins of free freshwater bodies, but they can be found from the few square centimeters of water accumulated in bromeliads to the millions of square kilometers of the surface of the oceans [1,2,3]. There are also some that live in different types of terrestrial environments located close to bodies of water or isolated from them [4].

*Spelaeometra* Polhemus & Ferreira, 2018 (Gerromorpha: Hydrometridae) is a monotypic and troglobitic genus, recently described based on material collected in the Peruaçu Caves region and adjacent areas, northern state of Minas Gerais, Brazil. Individuals of the genus display some of the most common troglomorphisms, such as reduced eyes, very elongated appendages, and light-colored body [5,6]. It was allocated in Hydrometridae for having the antennal segment IV with an apical invagination and specialized bristles, eyes located far from the anterior margin of the pronotum, the posterior pair of cephalic trichobothria being inserted in distinct tubercles, for lacking ocelli, and for lacking evidence of the opening of the metasternal and abdominal scent glands [7].

*Spelaeometra* can be distinguished from most other genera of the family mainly by its troglomorphisms and the complete absence of wings. When compared with the other troglobitic hydrometrid genus, *Cephalometra* Polhemus & Ferreira, 2018, it has a much shorter antenna, not reaching the apex of the body, and a more robust antennal segment I. Based on three specimens originating from the Gruna do Enfurnado Cave, municipality of Coribe, southwestern state of Bahia, Brazil, we present here the description of a new species of *Spelaeometra*, which is restricted to the cave habitat and therefore troglobitic.

## 2. Materials and Methods

The Gruna do Enfurnado Cave is located at the Serra do Ramalho karst area, municipality of Coribe, state of Bahia, Brazil. This karst area includes several caves and is subject to many threats, without any legal protection so far [8,9]. Serra do Ramalho is located in the southwestern portion of the state of Bahia, part of the Middle São Francisco River basin (Figure 1). According to the [10] classification, the climate is tropical dry (=semiarid), of “Aw” type, characterized by a dry winter (March to October) and an annual precipitation of around 640 mm [11]. The native regional vegetation consists of Caatinga (mesophytic and xeromorphic forests) interspersed with Cerrado (savannah-like).

Serra do Ramalho is dominated by a plateau formed by carbonatic (limestone) rocks of the Bambuí Group [8]. The Gruna do Enfurnado Cave has ca. 7600 m of passageways and about 3000 m of subterranean drainage [8,12]. It is one of the main caves in Serra do Ramalho, with a large population of a troglobitic catfish, *Rhamdia enfurnada* Bichuette & Trajano, 2005, and other troglobitic organisms [12].

The depth of the water body where our specimens were collected varied from 0.2 m to 4.0 m. The hydrometrids were found mainly in lentic reaches, with bottoms formed by silt, pebbles, and boulders, plus plant debris (Figure 2). Physicochemical water parameters measured (April 2017, end of the rainy season) were temperature = 20.9 °C; pH = 8.01; conductivity = 0.611 ms/cm; dissolved oxygen (DO) = 0.91 mg/L; salinity = 0.04%. The high pH and conductivity values are typical of karst waters. Large amounts of organic matter are present along the drainage due to floods during the rainy season, including large tree trunks. The aquatic cave fauna is highly dependent on allochthonous items carried from the surface mainly in this season.

Specimens were collected using small hand nets, fixed in 70% ethanol, and deposited in the Coleção Entomológica do Instituto Oswaldo Cruz, Rio de Janeiro, Brazil (CEIOC). Descriptions, photographs, and scanning electron micrographies were produced based on dry specimens. Abbreviations used for measurements are as follows: body length (BL), head length (HL), anteocular length (ANTL), maximum anteocular width (ANTWmax), minimum anteocular width (ANTWmin), transocular width (TOW), interocular width (IOW), postocular length (POSTL), maximum postocular width (POSTWmax), minimum postocular width (POSTWmin), clypeal length (CLL), basal clypeal width (BCLW), maximum clypeal width (CLWmax), length of antennal segments I–IV (ANT I, II, III, IV), length of labial segments I–IV (LB I–II, III, IV), ocular length (OL), maximum ocular width (EYE), pronotum length at midline (PL), pronotum width at anterior margin (PWAM), maximum pronotum width (PWmax), forecoxa/midcoxa distance (DIST1), midcoxa/hindcoxa distance (DIST2), femoral length (FEM), tibial length (TIB), length of tarsomeres I–III (TAR I, II, III), abdominal length (ABL), maximum abdominal width (ABWmax), minimum abdominal width (ABWmin), length of abdominal mediotergites I–VII (TERL I, II, III, IV, V, VI, VII), and maximum width of abdominal mediotergites I–VII (TERW I, II, III, IV, V, VI, VII). All measurements are given in millimeters.

For the micrographies, uncoated specimens were analyzed and illustrated with a Quanta 250 Scanning Electron Microscope (FEI Company, Hillsboro, OR, United States) in low-vacuum mode. Regular digital photographs were taken with a DFC 295 camera attached to a M205 C stereomicroscope with a Planapo 1.0× objective (Leica Camera AG, Solms, Germany). Figures were produced from stacks of images using LAS (Leica Application Suite) v3.7. The map was produced using QGIS Desktop 3.6.0 (QGIS Development Team, https://qgis.org/en/site/, accessed on 12 July 2023).

## 3. Results

*Spelaeometra hypogea* Cordeiro & Moreira, sp. nov. (Figure 3, Figure 4, Figure 5 and Figure 6)

urn:lsid:zoobank.org:act:A7248254-188B-4D29-9F36-6A853D62127F

Material examined. Holotype. BRAZIL–Bahia • Coribe, Gruna do Enfurnado Cave; 15 April 2010; M.E. Bichuette & J.E. Gallão leg.; 1 male, CEIOC 82865. Paratypes. Same data as holotype, 1 male, 1 female, CEIOC 82866.

Measurements. Apterous male holotype (paratype). BL: 2.85 (3.05), HL: 0.85 (0.86), ANTL: 0.55 (0.55), ANTWmax: 0.30 (0.34), ANTWmin: 0.30 (0.30), TOW: 0.31 (0.31), IOW: 0.28 (0.28), POSTL: 0.22 (0.24), POSTWmax: 0.30 (0.34), POSTWmin: 0.28 (0.32), CLL: 0.18 (0.20), BCLW: 0.06 (0.08), CLWmax: 0.10 (0.10), ANT: I: 0.35 (0.37), II: 0.34 (0.34), III: 1.24 (1.24), IV: 1.60 (1.60), LB: I: 0.12 (0.12), II: 0.08 (0.08), III: 1.54 (1.56), IV: 0.50 (0.57), OL: 0.06 (0.07), EYE: 0.01 (0.01), PL: 0.48 (0.48), PWAM: 0.32 (0.34), PWmax: 0.48 (0.50), DIST1: 0.30 (0.30), DIST2: 0.30 (0.30), Foreleg-FEM: 1.00 (1.05), TIB: 1.30 (1.30), TAR: I: 0.04 (0.04), II: 0.28 (0.28), III: 0.12 (0.13), Midleg-FEM: 1.20 (1.20), TIB: 1.44 (1.46), TAR: I: 0.04 (0.04), II: 0.28 (0.30), III: 0.12 (0.14), Hindleg-FEM: 1.54 (1.58), TIB: 2.25 (2.30), TAR: I: 0.05 (0.05), II: 0.42 (0.42), III: 0.14 (0.14), ABL: 1.60 (1.80), ABWmax: 0.44 (0.54), ABWmin: 0.28 (0.30), TERL: I: 0.20 (0.20), II: 0.20 (0.20), III: 0.20 (0.20), IV: 0.20 (0.22), V: 0.22 (0.24), VI: 0.24 (0.24), VII: 0.26 (0.30), TERW: I: 0.36 (0.40), II: 0.32 (0.34), III: 0.30 (0.32), IV: 0.30 (0.30), V: 0.26 (0.30), VI: 0.24 (0.28), VII: 0.20 (0.20).

Description. General color pale-yellow to yellowish-brown, with translucent appearance on appendages (Figure 3A). Head elongated, almost three times longer than wide; dorsal punctations concentrated medially (Figure 4B). Clypeus elongated, with truncated anterior margin and slightly emarginated lateral margins, almost hourglass-shaped; five long filiform setae distributed as an anterior pair, a single one centrally and a posterior pair (Figure 4B). Eye highly reduced, composed of just seven ommatidia, with two ocular setae (Figure 4D,E). A punctuation located dorsally to eye (Figure 4E). Antennal tubercle prominent in lateral view (Figure 4D). Antenna filiform; antennal segment I slightly arched; subequal in length to II; III and IV thinner than I and II; IV longest (Figure 3A). Dorsum of head with a pair of dorsal ridges rising near eyes and converging on the postocular region (Figure 4B). Posterior trichobothrium inserted on a tubercle, elongated, reaching anterior lobe of pronotum (Figure 4D,F). Ventral region of head flattened, with a pair of submedian longitudinal ridges (Figure 4C). Labium elongated, surpassing middle of abdomen; segments I and II short; segment III about three times as long as IV (Figure 4G,I). Pronotum subquadrate, with many punctures (Figure 4F); anterior and posterior lobes not well-defined. Pro- and mesopleura with punctations (Figure 4H). Pro-/meso- and meso-/metacetabula equidistant, without punctations (Figure 4H). Legs generally long, slender, except robust base of hind femur (Figure 3A and Figure 4A,I). Tarsi with three segments; tarsomere II longest. Claws well developed. Arolia absent. Abdomen 3.3 to 3.6 times longer than wide. Abdominal scent gland apparatus present, with scent orifice located at the middle of mediotergite IV (Figure 5). Abominal segment VIII cylindrical (Figure 4G and Figure 6A). Pygophore elongated, lateral margin with U-shape emargination (Figure 6A). Proctiger lacking basolateral and apical projections. Paramere large, twisted, facing mesally to genital capsule; hook-shaped apex facing mesal surface of the structure (Figure 6B).

Measurements. Apterous female. BL: 3.15, HL: 0.90, ANTL: 0.60, ANTWmax: 0.32, ANTWmin: 0.30, TOW: 0.32, IOW: 0.28, POSTL: 0.24, POSTWmax: 0.32, POSTWmin: 0.32, CLL: 0.18, BCLW: 0.08, CLWmax: 0.10, ANT: I: 0.34, II: 0.34, III: 1.25, IV: 1.55, LB: I: 0.12, II: 0.08, III: 1.50, IV: 0.75, OL: 0.07, EYE: 0.01, PL: 0.48, PWAM: 0.36, PWmax: 0.52, DIST1: 0.20, DIST2: 0.20, Foreleg-FEM: 0.95, TIB: 1.20, TAR: I: 0.04, II: 0.27, III: 0.12, Midleg-FEM: 1.22, TIB: 1.35, TAR: I: 0.05, II: 0.28, III: 0.13, Hindleg-FEM: 1.55, TIB: 2.30, TAR: I: 0.08, II: 0.40, III: 0.14, ABL: 1.70, ABWmax: 0.68, ABWmin: 0.24, TERL: I: 0.20, II: 0.20, III: 0.20, IV: 0.24, V: 0.24, VI: 0.24, VII: 0.14, TERW: I: 0.40, II: 0.34, III: 0.30, IV: 0.30, V: 0.26, VI: 0.24, VII: 0.14.

Description. Very similar to apterous male. General color light-brown. Anterior portion of head and distal half of femora yellowish-brown. Antennae, tibiae and tarsi pale-yellow (Figure 3B).

Etymology. The specific epithet hypogea (Latin, feminine adjective), meaning “underground”, refers to the cave habitat of this species.

## 4. Discussion

The authors of *Spelaeometra* [7] mentioned the lack of evidence of the scent orifice at abdominal mediotergite IV in this genus. However, we can observe the orifice in scanning electron microscopy images of *S. hypogea* Cordeiro & Moreira, sp. nov. (Figure 5). Such a structure is not observable under optical microscope (Figure 3) due to the coloration of the specimens and light diffraction, and is probably also present in *S. gruta*.

The new species described here resembles *S. gruta* Polhemus & Ferreira, 2018 in general shape and proportions of the body and appendages. *Spelaeometra hypogea* Cordeiro & Moreira, sp. nov., however, can be distinguished from it based on the following characteristics: (1) the general body color being pale-yellow to light-brown (Figure 3) (vs. orange to reddish-brown); (2) antennal segments I and II being subequal in length (vs. I longer than II); (3) the reduced gular lobe, not covering any labial segment (vs. covering segment I); (4) the male proctiger without apical projections (Figure 6A) (vs. with apical projections); and (5) the male paramere with convex dorsal margin after the curvature, with a narrower hook-like apex (Figure 6B) (vs. straight dorsal margin after the curvature, with wider subtriagular apex).

## 5. Conclusions

The Serra do Ramalho karst area is currently not protected by law, and the local biodiversity is subject to severe threats, including the extraction of the original vegetation and the development of mining projects. The Gruna do Enfurnado Cave harbors endemic troglobitic species, such as isopods, fishes, opilionids, and the new species described here. There is reason for major concern about the future of these endemic species and others not yet formally described. The proposition of a new conservation unit including the Gruna do Enfurnado Cave is a way to ensure their protection.

## Figures and Tables

**Figure 1 animals-13-03199-f001:**
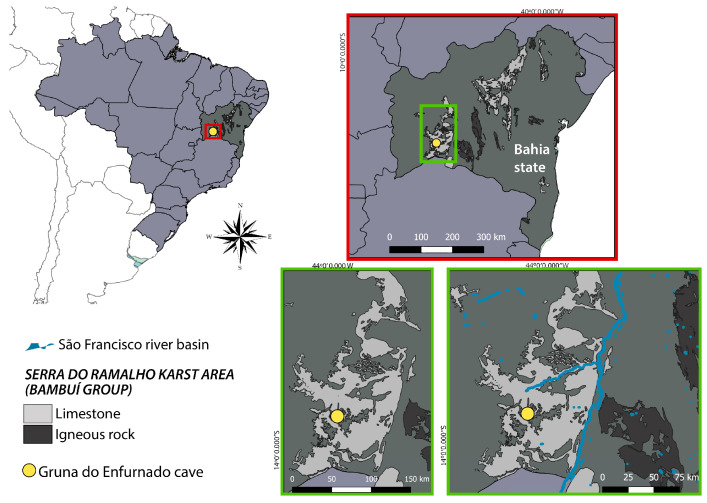
Geographic location of Gruna do Enfurnado Cave, Serra do Ramalho karst area, municipality of Coribe, state of Bahia, Brazil.

**Figure 2 animals-13-03199-f002:**
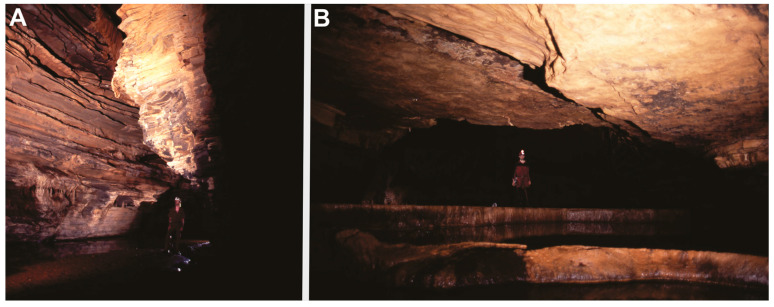
Habitat of *Spelaeometra hypogea* Cordeiro & Moreira, sp. nov. Gruna do Enfurnado Cave, state of Bahia, Brazil. (**A**) A river conduit in the aphotic zone, with bottom formed by silt. (**B**) Travertine pools with lentic waters in the aphotic zone.

**Figure 3 animals-13-03199-f003:**
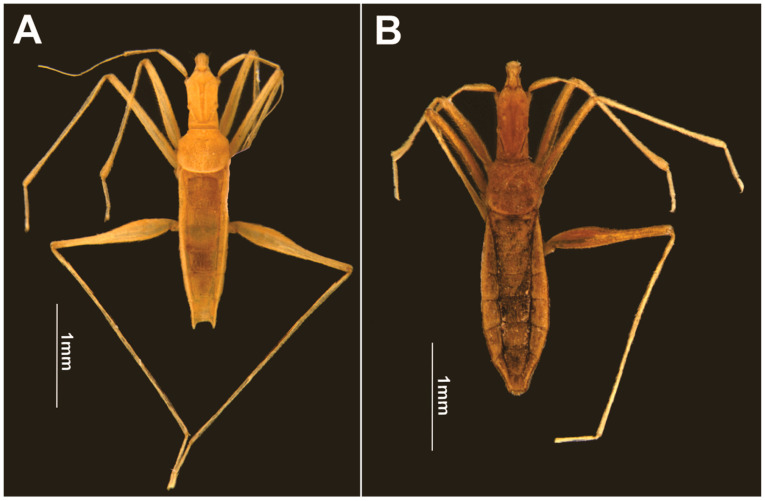
*Spelaeometra hypogea* Cordeiro & Moreira, sp. nov., habitus, dorsal view. (**A**) Male with genital capsule removed. (**B**) Female.

**Figure 4 animals-13-03199-f004:**
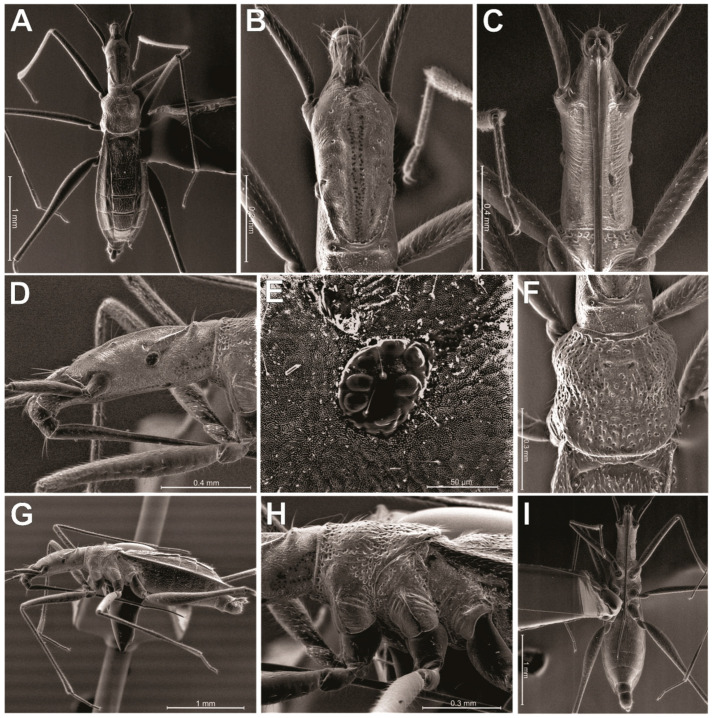
*Spelaeometra hypogea* Cordeiro & Moreira, sp. nov., scanning electron microscopy. (**A**) Male, habitus, dorsal view. (**B**) Head, dorsal view. (**C**) Head, ventral view. (**D**) Head, lateral view. (**E**) Reduced eye with seven ommatidia. (**F**) Pronotum. (**G**) Male, habitus, lateral view. (**H**) Thorax, lateral view. (**I**) Male, habitus, ventral view.

**Figure 5 animals-13-03199-f005:**
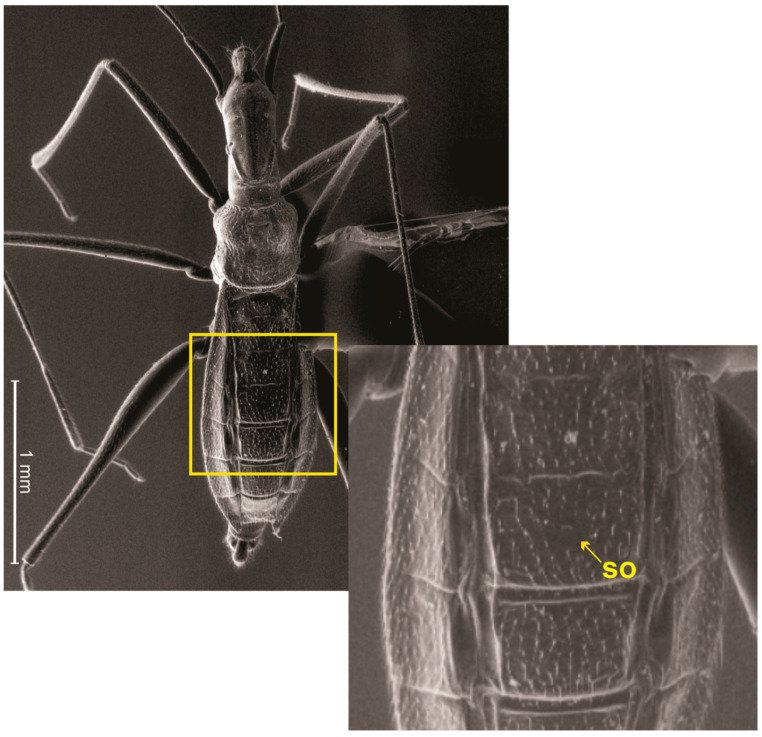
*Spelaeometra hypogea* Cordeiro & Moreira, sp. nov., scanning electron microscopy, male, habitus, dorsal view, and detail of abdominal mediotergites III–V (yellow square) showing scent orifice (SO) located at the middle of mediotergite IV.

**Figure 6 animals-13-03199-f006:**
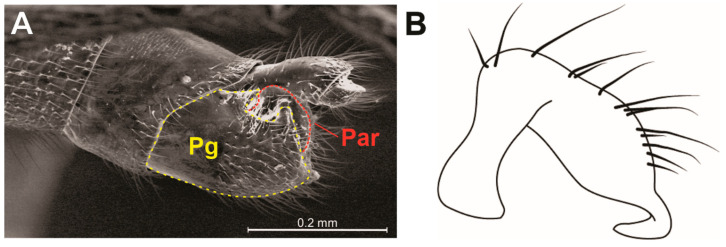
*Spelaeometra hypogea* Cordeiro & Moreira, sp. nov., male terminalia. (**A**) Genital capsule, lateral view, with pygophore (Pg) and paramere (Par) delimited by dotted lines. (**B**) Left paramere, lateral view.

## Data Availability

The data presented in this study are available in the article.

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
