# Peer review of "A New Species of Spelaeometra Polhemus & Ferreira, 2018 (Hemiptera: Heteroptera: Hydrometridae) from a Hotspot of Troglobites in Brazil, Serra do Ramalho Karst Area†"

_animals, 2023, doi:10.3390/ani13203199_

Round 1

Reviewer 1 Report

Dear Authors

The content was properly corrected and no new problems were found.

Author Response

The only comment provided by the reviewer was "Dear Authors The content was properly corrected and no new problems were found."

Therefore, there is no need for any corrections, according to this reviewer.

Reviewer 2 Report

The authors present the describing new species of Spelaeometra from a hotspot of troglobites in Brazil, Serra do Ramalho karst area, as the second species of this genus. This is a nice paper and I have no serious criticisms regarding methodology, results, and interpretation of results. The morphological descriptions are very concise and well-explanatory, and the photographs and illustrations are elaborate and aid in species identification. The literatures were well-surveyed. Only a few corrections concerning morphological terms are found, which are concerning the description part. Please check the PDF reviewed in detail.

Corrections

Line 24: antennomeres to antennal segments

Line 42: antennomeres to antennal segments

Line 49: antennomeres to antennal segments

Line 91: antennomeres to antennal segments

Line 92: articles to segments

Line 112: Insert “Measurements” as a headline.

Line 125: Insert “Description” as a headline.

Line 132: antennomeres to antennal segments

Line 137: articles to segments (or labial segments)

Line 138: articles to segments

Line 157: Insert “Measurements” as a headline.

Line 168: Etimology to Etymology

Line 189: antennomeres to antennal segments

Line 190: articles to segments

Line 190: articles to segments

Author Response

Line 24: antennomeres to antennal segments

Accepted

Line 42: antennomeres to antennal segments

Accepted

Line 49: antennomeres to antennal segments

Accepted

Line 91: antennomeres to antennal segments

Accepted

Line 92: articles to segments

Accepted

Line 112: Insert “Measurements” as a headline.

Accepted

Line 125: Insert “Description” as a headline.

Accepted

Line 132: antennomeres to antennal segments

Accepted

Line 137: articles to segments (or labial segments)

Accepted

Line 138: articles to segments

Accepted

Line 157: Insert “Measurements” as a headline.

Accepted

Line 168: Etimology to Etymology

Accepted

Line 189: antennomeres to antennal segments

Accepted

Line 190: articles to segments

Accepted

Line 190: articles to segments

Accepted

Reviewer 3 Report

The paper describes a new troglobiont water measurer from Brazil and adds thereby to the knowledge of this particular inhabitants to limestone cages, so far unknown from similar habitats in other parts of the world, and once again places Brazil as the center for diversification of the Hydrometridae. The paper is very well written and includes not only description of the apterous male and female, and a comparison with the congener, Spelaeometra gruta, but does also include ecological observations of the habitat. One could wish for a better comparison of the two species of Spelaeometra in which the diagnostic features of both types are not just compared in text but with actual pictures and drawings, but the illustrations in both papers are fortunately excellent and allows such comparisons easily. 

Author Response

All of the reviewer's comments were positive and they did not ask for anything in particular. Therefore, there is no need for corrections, according to this reviewer.